# Immunohistopathological Analysis of Immunoglobulin E-Positive Epidermal Dendritic Cells with House Dust Mite Antigens in Naturally Occurring Skin Lesions of Adult and Elderly Patients with Atopic Dermatitis

Ryoji Tanei [1,*] and Yasuko Hasegawa [2]

1 Department of Dermatology, Tokyo Metropolitan Geriatric Hospital and Institute of Gerontology, Itabashi, Tokyo 173-0015, Japan

2 Department of Geriatric Pathology, Tokyo Metropolitan Geriatric Hospital and Institute of Gerontology, Itabashi, Tokyo 173-0015, Japan; yhasegaw@tmig.or.jp

* Correspondence: rtanei@aol.com

**Abstract:** The immunopathogenic role of house dust mite (HDM) allergens in the development of skin lesions in atopic dermatitis (AD) has not yet been precisely clarified. We immunohistopathologically evaluated the localization of immunoglobulin E (IgE)-positive epidermal dendritic cells with HDM antigens in the skin lesions of patients with IgE-allergic AD. Using double-immunofluorescence and single-immunochemical staining methods, we analyzed biopsy specimens from the skin lesions of six patients with IgE-allergic AD and HDM allergy and 11 control subjects with inflammatory skin disorders. Inflammatory dendritic epidermal cells (IDECs; CD11c+ and CD206+ cells) were markedly observed in the central area of the spongiotic epidermis of skin lesions in all AD patients. Furthermore, IgE-positive IDECs with HDM antigens in the central areas of the spongiosis were found in four of the six (66.7%) AD patients. Langerhans cells (LCs; CD207+ cells) with HDM antigens were also observed in the peripheral areas of the spongiosis. Infiltration of CD4+ and CD8+ T cells in association with IgE-positive IDECs and LCs with HDM antigens was seen in the spongiotic epidermis. An IgE-mediated delayed-type hypersensitivity reaction, in combination with IgE-bearing dendritic cells, specific T cells, keratinocytes, and HDM antigens, may lead to spongiotic tissue formation in eczematous dermatitis in AD.

**Keywords:** double-immunofluorescence staining; house dust mite antigens; immunohistopathology; inflammatory dendritic epidermal cells; Langerhans cells; spongiosis

## 1. Introduction

Although the pathophysiology of atopic dermatitis (AD) is complex, its basic axis is considered to be T cell-mediated immune reactions in association with immunoglobulin (Ig)E-type hypersensitivity specific to environmental exposures, such as to aeroallergens and food allergens [1–3]. Epidermal and dermal dendritic cells (DCs), such as Langerhans cells (LCs), inflammatory dendritic epidermal cells (IDECs), and dermal inflammatory DCs, are thought to be the principal antigen-presenting cells and potential sources of inflammatory mediators in AD that initiate and sustain the cellular infiltration of allergen-specific T cells and other effector and regulatory cells. LCs are defined as epidermal DCs that contain Birbeck granules and mainly express cluster of differentiation (CD)1a and CD207 (langerin) antigens. IDECs are defined as epidermal inflammatory DCs that do not contain Birbeck granules and mainly express CD1a, CD11b, CD11c, and CD206 antigens [4,5]. Dermal inflammatory DCs (so-called atopic DCs) that upregulate T helper (Th)2-attracting chemokines (e.g., thymus and activation-regulated chemokine (TARC)/CC Chemokine Ligand (CCL) 17) express CD11c antigens [5]. LCs, IDECs, and dermal inflammatory DCs are all classified as subsets of myeloid DCs. Another subset of DCs, plasmacytoid

DCs (CD11c-DCs that express CD123 antigens) are known to play an important role in innate immunity against viral infection by producing interferon-alpha in the dermis of AD lesions [5–7]. LCs, IDECs, dermal inflammatory DCs, and plasmacytoid DCs have major histocompatibility complex class I and II molecules and can become immunoglobulin (Ig) E-bearing DCs with complexes of IgE and high-affinity receptor FcεRI in the major form of AD with IgE allergy (IgE-allergic (extrinsic form) AD) [1,3–6]. It has been considered that IgE-bearing myeloid DCs can capture large amounts of allergens and may cause IgE-mediated delayed-type hypersensitivity reactions in AD [1,8]. However, the immunopathogenic roles of common environmental allergens, such as house dust mite (HDM) allergens, in the development of skin lesions in IgE-allergic AD have not yet been precisely clarified [3], although previous studies using animal models have indicated that eczematous skin reactions could possibly be induced by IgE-allergic sensitization [9] and that HDM allergens can promote an IgE-allergic AD phenotype in a mouse model [10].

In the present study, we analyzed the localization of IgE-bearing DCs, mainly LCs and IDECs, in association with HDM antigens from Dermatophagoides species in chronic active skin lesions of adults and elderly patients with AD and allergic sensitization to HDMs. We also discuss the possible role of HDM antigens in the pathomechanism of eczematous dermatitis in IgE-allergic AD.

## 2. Materials and Methods

### 2.1. Subjects and Skin Samples

Skin biopsy specimens from six Japanese patients (Cases 1 to 6; 4 elderly and 2 adult patients) with IgE-allergic AD and high titers of serum-specific IgEs against HDMs (two Dermatophagoides species: D. farinae and D. pteronyssinus) were analyzed by immuno-histopathological methods. As controls, skin biopsy specimens from six patients (Cases 7 to 12; six elderly patients) with non-eczematous inflammatory skin disorders and serum hyper-IgE and five patients (Cases 13 to 17; 4 elderly and 1 adult patients) with inflammatory skin disorders and spongiotic tissue within the epidermis were also analyzed. These subjects were selected to confirm that the localization of IgE-positive DCs and HDM antigens in eczematous dermatitis of IgE-allergic AD could be characteristic findings of AD compared to other inflammatory skin disorders. We did not add healthy participants to the control subjects, because our preliminary analysis did not reveal any specific findings in their skin. In this study, patients over 60 years of age were considered to be elderly. The skin specimens were obtained from chronic skin lesions (lichenified eczema) of the patients with AD and from lesioned skin of the control subjects. All patients with AD fulfilled the diagnostic clinical criteria of Hanifin and Rajka [11], and the severity of AD was scored according to the Eczema Area and Severity Index (maximum, 72) [12]. AD patients and the control subjects all provided written informed consent for each biopsy and for the use of their specimens for research. The study protocols were approved by the Ethics Committee of the Tokyo Metropolitan Geriatric Hospital and Institute of Gerontology (No. R15-42).

### 2.2. Histological, Immunohistochemical, and Double-Immunofluorescence Staining

Formalin-fixed paraffin-embedded 3-μm-thick sections were used for routine hematoxylin-eosin staining and single-immunohistochemical staining. Frozen 7-μm-thick sections were used for double-immunofluorescence staining. The single-immunohistochemical staining and double-immunofluorescence staining were performed with the same procedure used in our previous studies [13,14].

The following primary monoclonal antibodies (mAbs) and polyclonal antibodies (pAbs) were used: mouse mAbs against CD4 (helper/inducer/regulatory T cells, #713181; Nichirei, Tokyo, Japan), CD8 (cytotoxic T cells, #713201; Nichirei), CD206 (IDECS [4,5]/macrophage, MCA2519; Bio-Rad, Hercules, CA, USA), CD207 (LCs, ABIN1027332; antibodies-online Inc., Atlanta, GA, USA), mast cell tryptase (mast cells, AA1; Abcam, Tokyo, Japan), and IgE ε-chain (IgE, MH25-1; Santa Cruz Biotechnology, Santa Cruz, CA, USA); rabbit mAbs against CD3 (T cells, #713591; Nichirei) and CD11c

(dermal DCs/IDECS [5,7], EP1347Y; LSBio, Seattle, WA, USA); and rabbit pAbs against CD206 (IDECS/macrophage, 18704-1-AP; Proteintech, Rosemont, IL, USA), D. farinae 1 (Der f1; the main allergenic component of D. farinae that can cross-react with D. pteronyssinus 1; Mite Der f1, LB-7111; Cosmo Bio, Tokyo, Japan), and Mite Extract (Mite Crude Extract containing antigens from D. farinae and D. pteronyssinus; LB-5199; Cosmo Bio).

The dilution ratios for the primary antibodies were: 1:800 for anti-CD206, 1:200 for anti-IgE (mouse mAb), 1:250 for anti-CD11c (rabbit mAb), 1:4000 for anti-CD206, 1:1000 for anti-Derf1, and 1:500 for anti-Mite Extract (rabbit pAb). The other primary antibodies did not need to be diluted before use.

Double-immunofluorescence staining was performed with a pair of mouse and rabbit mAbs, or a pair of a mouse mAb and a rabbit pAb by using two suitable primary antibodies for double-staining conditions. The combined secondary antibodies were biotinylated anti-mouse IgG (BA-9200; Vector Laboratories, Burlingame, CA, USA) and biotinylated anti-rabbit IgG (BA-1000; Vector Laboratories). The streptavidin-fluorescein conjugates used were DyLight488 streptavidin (SA-5488; Vector Laboratories) and DyLight594 streptavidin (SA-5594; Vector Laboratories). Nuclei were labeled with $4'$,6-diamidino-2-phenylindole. Double-immunofluorescence-stained specimens were observed under fluorescence microscopy (BIOREVO BZ-9000; Keyence, Osaka, Japan).

### 2.3. Evaluation of Immunohistopathological Findings

The analyses were performed by qualitative and quantitative evaluations of the immunohistopathological findings. In the quantitative evaluations, the numbers of immunopositive cells in a 0.24 mm$^2$ area of the epidermis or dermis were counted under a microscope at 200× magnification. Cell numbers were counted in the most cell-rich area from among three or more observed areas in the non-spongiotic epidermis or superficial dermis in all cases and from one or more observed areas in the spongiotic epidermis in cases of AD and inflammatory skin disorders with spongiotic tissue within the epidermis. Only cells that were clearly positive for the immune-marker antigen of interest were counted. In this study, for the analysis of DCs in the epidermis, we used both CD11c and CD206 as markers of IDECs, and we used CD207 as a marker of LCs [4,5,7]. For the quantitative evaluations of HDM antigens, we mainly analyzed Der f1, because it is well known to be the major allergen component of HDMs [15]. For statistical analysis of the quantitative data, the Welch test was used. Values of $p < 0.05$ were considered to be statistically significant. We performed statistical comparisons mainly for the items associated with AD subjects and the control subjects with non-eczematous inflammatory skin disorders and serum hyper-IgE, since the present study primarily focused on the role of IgEs against HDM antigens. Data analysis was performed using EZR software version 1.54 (Saitama Medical Center Jichi Medical University, Saitama, Japan) [16].

### 3. Results

#### 3.1. Clinical and Laboratory Data of the Patients with AD and the Control Subjects

Clinical and laboratory data of the AD patients and control subjects are summarized in Table 1. In AD cases and the control cases with non-eczematous inflammatory skin disorders, no statistically significant difference was found in mean value ± standard deviation of serum total IgE (AD cases: 33,822.8 ± 45,077.2 IU/mL, control cases: 27,982.2 ± 49,632.9 IU/mL; $p = 0.835$).

**Table 1.** Clinical and laboratory data of patients with atopic dermatitis and the control subjects.

| Case | Age/Sex | Disease | Onset [Approximate Age] | Biopsy Site | Topical Steroid Treatment [†] | Serum Total IgE (IU/mL) | Allergen-Specific IgEs against HDMs [‡] | EASI |
|---|---|---|---|---|---|---|---|---|
| 1 | 78/M | AD | Late adulthood [47 y.o.] | Forearm | Strong 2 months | 28,715 | High titer against Der f and Der p | 42.1 |
| 2 | 61/F | AD | Adolescence [15 y.o.] | Forearm | Not used | 10,198 | High titer against Der f and Der p | 29.4 |
| 3 | 83/F | AD | Childhood [Elementary school age] | Thigh | Mild 4 months | 11,571 | High titer against Der f [§] | 33.5 |
| 4 | 49/M | AD | Early childhood [2 to 3 y.o.] | Abdomen | Not used | 124,515 | High titer against Der f and Der p | 54.5 |
| 5 | 84/M | AD | Late adulthood [50 y.o.] | Upper back | Not used | 19,757 | High titer against Der f and Der p | 15.5 |
| 6 | 40/M | AD | Early childhood [6 y.o.] | Chest | Very strong 2 months | 8181 | High titer against Der f and Der p | 33.8 |
| 7 | 78/M | EGPA | Elderly | Lower leg | Not used | 3007 | Doubtful reaction against Der f [§] | NA |
| 8 | 83/M | DH | Elderly | Back | Very strong 0.5 month | 3101 | Negative [§] | NA |
| 9 | 75/M | BP | Elderly | Buttock | Very strong 2 months | 6235 | Negative [§] | NA |
| 10 | 84/M | BP | Elderly | Abdomen | Weak 0.5 month | 14,195 | Negative [§] | NA |
| 11 | 80/M | BP | Elderly | Abdomen | Strongest 1 month | 128,839 | Moderate titer against Der f [§] | NA |
| 12 | 85/M | MPE | Elderly | Abdomen | Very strong 3 months | 12,516 | Negative [§] | NA |
| 13 | 73/M | NE | Elderly | Lower leg | Strong 2 months | 25 | Negative [§] | 9.6 |
| 14 | 59/M | ICD | Late adulthood | Forearm | Not used | 46 | Low titer against Der f [§] | 0.8 |
| 15 | 83/F | EE | Elderly | Back | Not used | 97 | Negative [§] | 21.6 |
| 16 | 82/M | EE | Elderly | Forearm | Not used | 1429 | Negative [§] | 33.9 |
| 17 | 77/M | BP with ES | Elderly | Sole | Not used | 469 | NT | NA |

[†] Strength of topical corticosteroids and the treatment periods before skin biopsy. [‡] Determined using MAST (BML, Tokyo, Japan), View Allergy (LSI Medience, Tokyo, Japan), and/or the CAP-FEIA system (BML, Tokyo, Japan): High titer, class 6 or 5; Moderate titer, class 4 or 3; Low titer, class 2; Doubtful reaction, class 1; and Negative, class 0. [§] Not tested for Der p. AD, atopic dermatitis; BP, bullous pemphigoid; Der f, *Dermatophagoides farinae*; Der p, *Dermatophagoides pteronyssinus*; DH, dermatitis herpetiformis; EASI, Eczema Area and Severity Index (maximum, 72); EE, eczematous erythroderma; EGPA, eosinophilic granulomatosis with polyangiitis; ES, eosinophilic spongiosis; F, female; HDM, house dust mite; ICD, irritant contact dermatitis; IgE, immunoglobulin E; M, male; MPE, maculopapular exanthema; NA, not available; NE, nummular eczema; NT, not tested; y.o., years old.

### 3.2. Double-Immunofluorescence and Immunohistochemical Studies

Results of the double-immunofluorescence and immunohistochemical studies are summarized in Table 2.

**Table 2.** Double-immunofluorescence and immunohistochemical staining results for the main analyses of infiltrating cells with immune markers for IgE, dendritic cells, and house dust mite antigens in skin lesions of atopic dermatitis patients and the control subjects.

| Category | | | | | | | | | | | | | | |
|---|---|---|---|---|---|---|---|---|---|---|---|---|---|---|
| Case | Disease | Double-Immunofluorescence [†] | | | | | | | | | | | Immunohistochemistry [‡] | |
| | | Epidermis | | | | | | | | Dermis | | | Epidermis | | Dermis |
| | | IgE+ CD11c+ Cells | IgE+ CD206+ Cells | IgE− CD206+ Cells | IgE+ Der f1+ Cells | CD206+ Der f1+ Cells | CD207+ Der f1+ Cells | IgE+ Mite ext.+ Cells | CD206+ Mite ext.+ Cells | IgE+ CD11c+ Cells | IgE+ Der f1+ Cells | IgE− Der f1+ Cells | CD11c+ Cells | CD207+ Cells | CD11c+ Cells |
| **AD** | | | | | | | | | | | | | | | |
| 1 | AD | 43 | 28 | 6 | 13 | 10 | 10 | 11 | 9 | 44 | 9 | 11 | 45 | 46 | 83 |
| 2 | AD | 50 | 52 | 1 | 11 | 7 | 3 | 8 | 23 | 60 | 14 | 16 | 35 | 26 | 96 |
| 3 | AD | 12 | 21 | 3 | 5 | 1 | 3 | 5 | 2 | 41 | 14 | 18 | 42 | 30 | 85 |
| 4 | AD | 23 | 6 | 1 | 0 | 2 | 0 | 0 | 0 | 70 | 5 | 4 | 45 | 45 | 68 |
| 5 | AD | 11 | 11 | 0 | 23 | 8 | 14 | NA | NA | 40 | 12 | 15 | 6 | 27 | 22 |
| 6 | AD | 21 | 15 | 1 | 6 | 8 | 2 | 8 | 12 | 44 | 15 | 14 | 11 | 10 | 60 |
| Mean (±SD) [¶] | | 26.7 (±16.2) *,** | 22.2 (±16.5) *,** | 2.0 (±2.2) | 9.7 (±8.0) *,** | 6.0 (±3.6) | 5.3 (±5.4) | 6.4 (±4.2) [§] *,** | 9.2 (±9.1) [§] | 49.8 (±12.3) *,** | 11.5 (±3.8) *,** | 13 (±5.0) *,** | 30.7 (±17.6) * | 30.7 (±13.4) | 69 (±26.4) |
| **Non-eczematous inflammatory skin disorders with serum hyper-IgE** | | | | | | | | | | | | | | | |
| 7 | EGPA | 0 | 0 | 0 | 0 | 0 | 0 | 0 | 0 | 1 | 0 | 0 | 6 | 20 | 39 |
| 8 | DH | 0 | 0 | 0 | 0 | 1 | 0 | 0 | 0 | 3 | 0 | 0 | 1 | 26 | 61 |
| 9 | BP | 3 | 2 | 0 | 0 | 0 | 0 | 0 | 0 | 3 | 2 | 4 | 0 | 13 | 46 |
| 10 | BP | 0 | 0 | 0 | 0 | 0 | 2 | 0 | 0 | 4 | 0 | 0 | 1 | 9 | 68 |
| 11 | BP | 1 | 1 | 1 | 0 | 0 | 8 | 0 | 0 | 11 | 0 | 4 | 3 | 31 | 30 |
| 12 | MPE | 0 | 0 | 0 | 0 | 0 | 0 | 0 | 0 | 9 | 4 | 4 | 1 | 36 | 38 |
| Mean (±SD) [¶] | | 0.7 (±1.2) | 0.5 (±0.8) | 0.2 (±0.4) | 0 (±0) | 0.2 (±0.4) | 1.7 (±3.2) | 0 (±0) | 0 (±0) | 5.2 (±3.9) | 1 (±1.7) | 2 (±2.2) | 2 (±2.2) | 22.5 (±10.4) | 47 (±14.6) |
| **Inflammatory skin disorders with spongiotic tissue within the epidermis** | | | | | | | | | | | | | | | |
| 13 | NE | 0 | 0 | 54 | 0 | 0 | 0 | 0 | 1 | 2 | 0 | 0 | 78 | 12 | 81 |
| 14 | ICD | 0 | 0 | 25 | 0 | 0 | 2 | 0 | 0 | 0 | 0 | 1 | 37 | 16 | 82 |
| 15 | EE | 0 | 0 | 15 | 0 | 0 | 4 | 0 | 0 | 0 | 0 | 0 | 22 | 43 | 130 |
| 16 | EE | 0 | 0 | 11 | 0 | 0 | 1 | 1 | 0 | 2 | 0 | 0 | 21 | 40 | 56 |
| 17 | BP with ES | 0 | 0 | 12 | 0 | 0 | 2 | 0 | 0 | 0 | 0 | 0 | 66 | 28 | 49 |
| Mean (±SD) [¶] | | 0 (±0) | 0 (±0) | 23.4 (±18.0) | 0 (±0) | 0 (±0) | 1.8 (±1.5) | 0.2 (±0.4) | 0.2 (±0.4) | 0.8 (±1.1) | 0 (±0) | 0.2 (±0.4) | 44.8 (±25.9) | 27.8 (±13.9) | 79.6 (±31.8) |

[†] Using frozen sections. The cells that the immune markers indicate in the epidermis: IgE+ CD11c+ cells are IgE-bearing IDECs; IgE+ CD206+ cells are IgE-bearing IDECs; IgE− CD206+ cells are IDECs without IgE expression; IgE+ Der f1+ cells are IgE-bearing IDECs and/or LCs that co-localized with Der f1; CD206+ Der f1+ cells are IDECs that co-localized with Der f1; IgE+ Mite ext.+ cells are IgE-bearing IDECs and/or LCs that co-localized with Mite ext.; and CD206+ Mite ext.+ cells are IDECs that co-localized with Mite ext. The cells that the immune markers indicate in the upper dermis: IgE+ CD11c+ cells are IgE-bearing dermal inflammatory DCs and IgE+ Der f1+ and IgE+ Mite ext.+ cells are IgE-bearing infiltrating cells (e.g., DCs and mast cells) that co-localized with Der f1 or Mite ext. [‡] Using paraffin sections. The cells that the immune markers indicate: CD11c+ cells are IDECs; CD207+ cells are LCs in the epidermis; and CD11c+ cells are dermal DCs in the upper dermis. [¶] Data are presented as the mean ± standard deviation. * $p < 0.05$, compared to non-eczematous inflammatory skin disorders with serum hyper-IgE. ** $p < 0.05$, compared to inflammatory skin disorders with spongiotic tissue within the epidermis. [§] Data from the 5 cases. AD, atopic dermatitis; BP, bullous pemphigoid; CD, cluster of differentiation; DCs, dendritic cells; Der f1, Dermatophagoides farinae 1; EE, eczematous erythroderma; EGPA, eosinophilic granulomatosis with polyangiitis; ES, eosinophilic spongiosis; DH, dermatitis herpetiformis; ICD, irritant contact dermatitis; IDECs, inflammatory dendritic epidermal cells; IgE, immunoglobulin E; LCs, Langerhans cells; Mite ext., Mite Extract antigens; MPE, maculopapular exanthema; NE, nummular eczema.

### 3.3. Hematoxylin-Eosin Staining

Hematoxylin-eosin-stained sections of AD cases showed chronic eczematous reactions with epidermal hyperplasia, mononuclear cell infiltration with a few eosinophils, and increased infiltration of mast cells in the upper dermis in all cases, and obvious spongiosis (Figure 1a: case 1) was observed in the epidermis in four (case 1 to 4) of the six cases. Among the control cases, characteristics of each disease were observed, and no spongiotic epidermis was seen in the cases with non-eczematous inflammatory skin disorders and serum hyper-IgE.

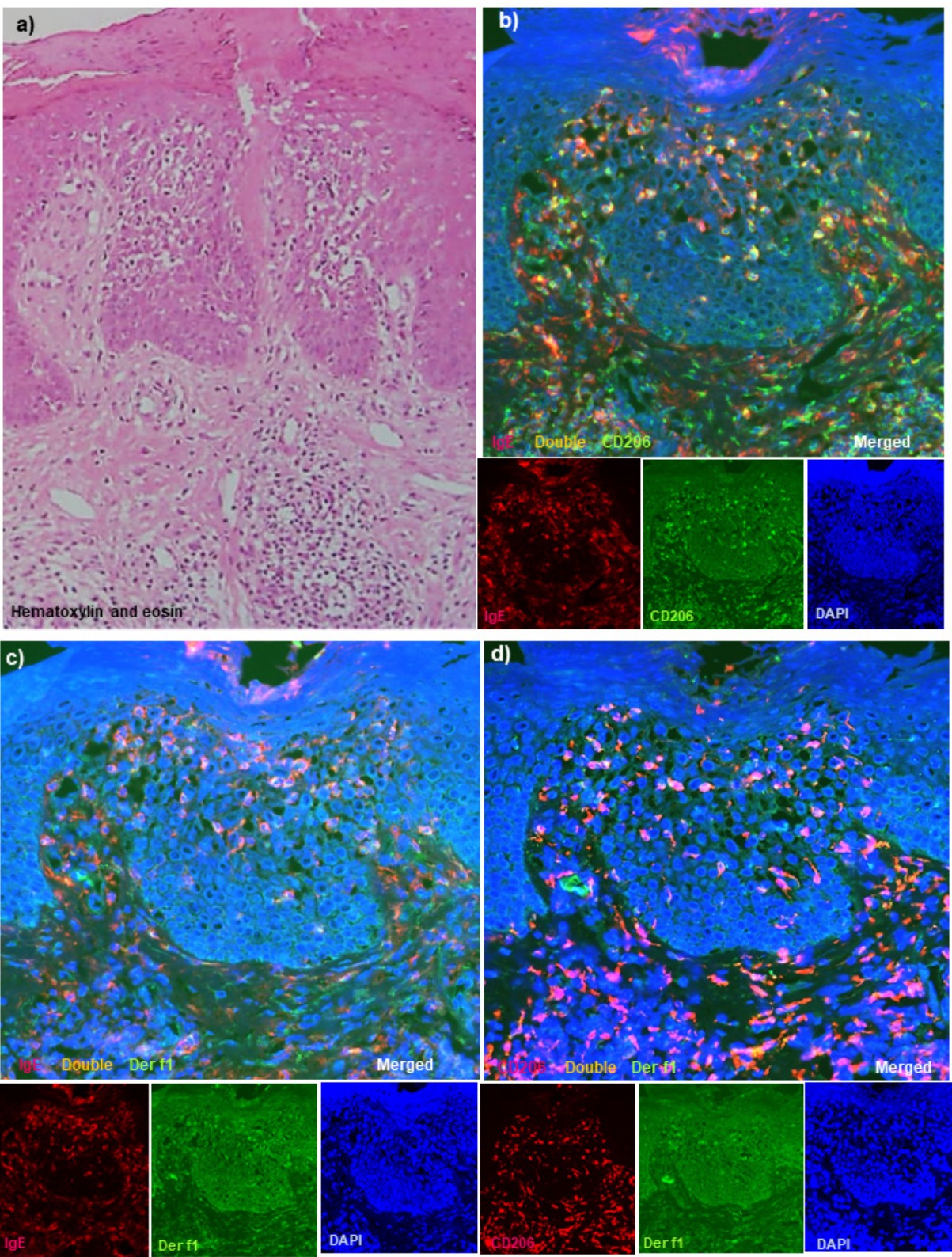

**Figure 1.** *Cont.*

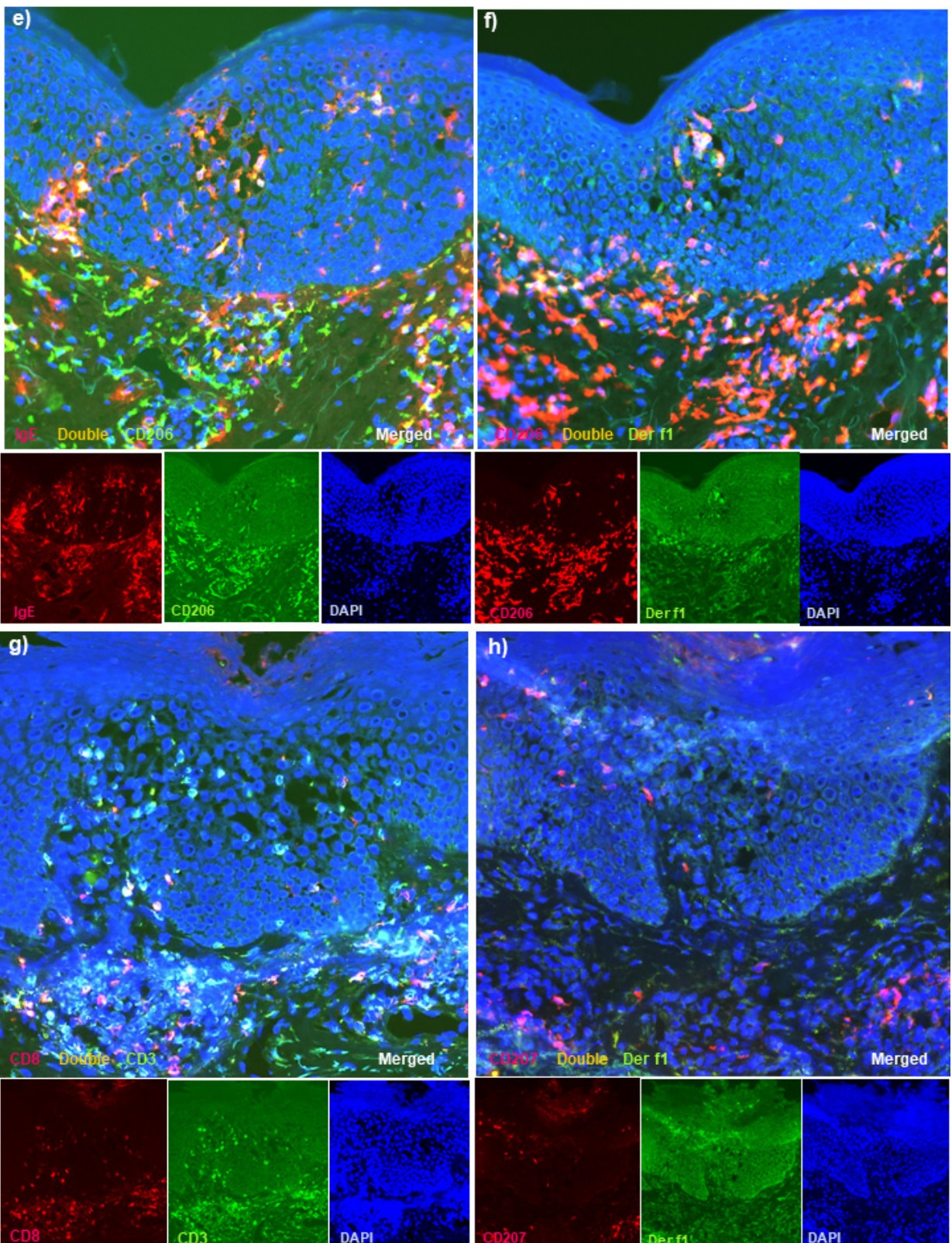

**Figure 1.** Routine and double-immunofluorescence staining for active lesions of chronic AD in patients with IgE-allergic AD and HDM allergy. (**a**) Chronic eczematous reactions with a scaly crust, epidermal hyperplasia, and obvious spongiosis with mononuclear cell infiltration were observed in the epidermis. (**b**) Double-positive IgE+ CD206+ cells (yellow) markedly infiltrated into the spongiotic epidermis of the active lesions of chronic AD. (**c**) Double-positive IgE+ Der f1+ cells (yellow)

were observed among the infiltrating cells in the same spongiotic epidermis. (**d**) Double-positive CD206+ Der f1+ cells (yellow) were also found among the infiltrating cells in the same spongiotic epidermis. (**e**) Double-positive IgE+ CD206+ cells (yellow) were observed in the spongiotic epidermis of active lesions of chronic AD. (**f**) Double-positive CD206+ Der f1+ cells (yellow) were found among the infiltrating cells in the same spongiotic epidermis. (**g**) Both double-positive CD3+ CD8+ cells (yellow) and single-positive CD3+ CD8−cells (green) were also observed in the spongiotic epidermis. (**h**) CD207+ cells (red) were less localized to the central area, and some double-positive CD207+ Der f1+ cells (yellow) were present in the peripheral area of the spongiotic epidermis. (**a**): Case 1; hematoxylin-eosin staining. (**b–d,g,h**): Case 1; double-immunofluorescence staining. (**e,f**): Case 6; double-immunofluorescence staining. In the double-immunofluorescence staining, nuclei were labeled with 4′,6-diamidino-2-phenylindole (DAPI; blue). Sets of figures, i.e., (**b–h**), represent serial sections. Original magnifications: 100×, (**a**); 200×, (**b–h**).

### 3.4. Double-Immunofluorescence Staining

Using serial frozen sections, we confirmed the presence of various degrees (from mild to the severe vesicular form) of spongiosis in the epidermis of the skin lesions in all AD cases. In the sections of double-immunofluorescence staining, a few infiltrating double-positive IgE+ CD11c+ cells and double-positive IgE+ CD206+ cells (i.e., IgE-bearing IDECs) were seen scattered in the middle to lower epidermis in the non-spongiotic epidermis, while they were observed to aggregate in higher numbers in the central areas of the spongiotic epidermis (Figure 1b: case 1). In addition, double-positive IgE+ Der f1+ cells and double-positive CD206+ Der f1+ cells showing the same localization as the immunostained Der f1 antigen-loaded IgE-bearing IDECs were also observed in the central areas of the spongiotic epidermis in four (cases 1, 2, 5, and 6) of the six cases (Figure 1b–d: case 1; Figure 1e,f: case 6). These immuno-positive cells were accompanied by infiltrating CD3+ CD8+ cells (i.e., cytotoxic T cells) and CD3+ CD8−(CD4+) cells (i.e., helper or regulatory T cells) in the spongiotic epidermis (Figure 1g: case 1). In contrast to the CD11c+ cells and CD206+ cells (i.e., IDECs) that aggregated in the central areas of the spongiotic epidermis, CD207+ cells (i.e., LCs) were observed mainly in the peripheral areas of the spongiotic epidermis, and some double-positive CD207+ Der f1+ cells (i.e., Der f1 antigen-loaded LCs) were observed in those CD207+ cells (Figure 1h: case 1). On the other hand, double-positive CD207+ Der f1+ cells were also observed in areas of the non-spongiotic epidermis (e.g., perifollicular epidermis), and some of them showed the same distribution of the double-positive IgE+ Der f1+ cells in those areas; however, they were accompanied by almost no infiltrating CD3+ T cells.

Using another antibody against HDM antigens (against Mite Extract antigens), we also confirmed the presence of double-positive cells for IgE and Mite Extract antigens, and double-positive cells for CD206 and Mite Extract antigens with the same localization as the double-positive IgE+ CD11c+ cells and IgE+ CD206+ cells (i.e., Mite Extract antigen-loaded IgE-bearing IDECs) in the central area of the spongiotic epidermis in three (cases 1, 2, and 6) of the five cases (Figure 2a–d: case 2). Unfortunately, one case (case 5) could not be examined for Mite Extract antigens, because the tissue section with spongiotic epidermis had been exhausted.

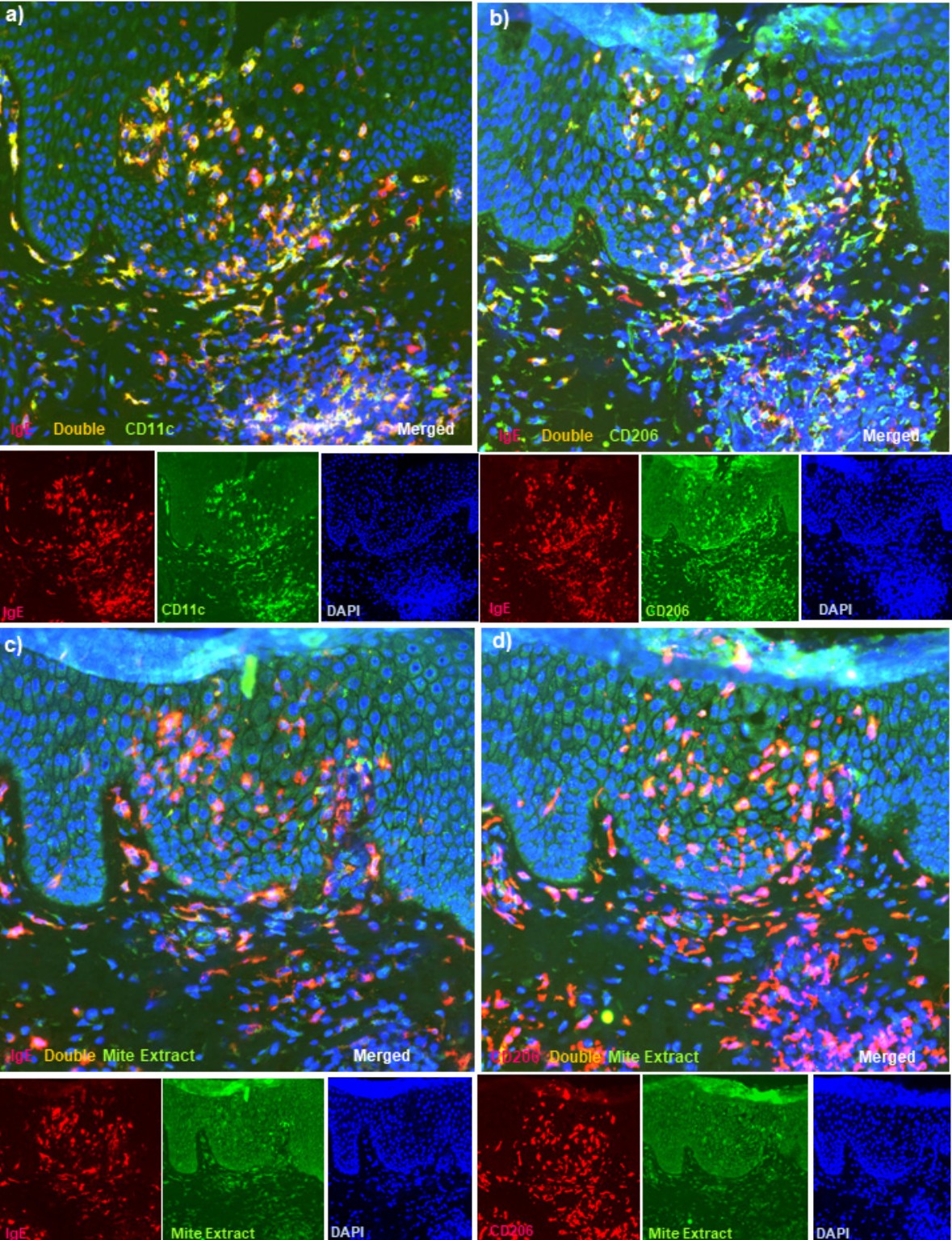

**Figure 2.** Double-immunofluorescence staining for active lesions of chronic AD in patients with IgE-allergic AD and HDM allergy. (**a**) Double-positive IgE+ CD11c+ cells (yellow) markedly infiltrated into the spongiotic epidermis of active lesions of chronic AD. (**b**) Double-positive IgE+ CD206+ cells (yellow) were also observed in the spongiotic epidermis. Note: almost all CD206+ cells in the spongiotic epidermis were IgE-positive. (**c**) Double-positive IgE+ Mite Extract antigen+ cells (yellow) were observed among the infiltrating cells in the same spongiotic epidermis. (**d**) Double-positive CD206+ Mite Extract antigen+ cells (yellow) were also observed among the infiltrating cells in the same spongiotic epidermis. (**a–d**): Case 2; double-immunofluorescence staining. Sets of (**a–d**) represent serial sections. Original magnifications: 200×, (**a–d**).

Regarding cell infiltration of the upper dermis in AD cases, infiltrating double-positive IgE+ CD11c+ cells (i.e., IgE-bearing dermal inflammatory DCs) were observed mainly in the papillary and subpapillary dermis in all of the AD cases. A small number of infiltrating double-positive IgE+ Der f1+ cells (i.e., Der f1 antigen-loaded IgE-bearing cells) were also observed in the papillary and subpapillary dermis.

In the control cases with non-eczematous inflammatory skin disorders and serum hyper-IgE, infiltration of only a small number of double-positive IgE+ CD11c+ cells and IgE+ CD206+ cells in the epidermis (i.e., IgE-bearing IDECs) and double-positive IgE+ CD11c+ cells in the upper dermis (i.e., IgE-bearing dermal inflammatory DCs) was seen. Furthermore, infiltrating double-positive cells with HDM antigens (i.e., Der f1 antigens and Mite Extract antigens) were scarcely found in the epidermis by double-immunofluorescence staining for IgE and Der f1, IgE and Mite Extract antigens, CD206 and Der f1, and CD206 and Mite Extract antigens; in addition, they were rarely found in the upper dermis by the staining for IgE and Der f1 (Figure 3a,b: case 11). In the control cases with inflammatory skin disorders and spongiotic tissue within the epidermis, infiltrating double-positive cells with HDM antigens were scarcely found in the spongiotic epidermis by double-immunofluorescence staining for IgE and Der f1, IgE and Mite Extract antigens, CD206 and Der f1, and CD206 and Mite Extract antigens (Figure 3c,d: case 13).

In the comparisons between the AD and control cases, the mean numbers of double-positive cells, i.e., IgE+ CD11c+ cells, IgE+ CD206+ cells, IgE+ Der f1+ cells, CD206+ Der f1+ cells, and IgE+ Mite Extract+ cells in the epidermis and IgE+ CD11c+ cells and IgE+ Der f1+ cells in the upper dermis, in the AD cases were significantly higher than those in each of the control groups. The mean numbers of single-positive (IgE−) Der f1+ cells in the upper dermis in AD cases were also significantly higher than those in each of the control groups (Table 2).

*3.5. Single-Immunohistochemical Staining*

Using serial paraffin-embedded immunostained sections, we confirmed the presence of various degrees of spongiosis in five (cases 1 to 5) of the six AD cases. Serial sections of hematoxylin-eosin staining and immunostaining demonstrated the infiltration of CD11c+ cells into the spongiotic epidermis in the 5 cases, and infiltrating IgE+ cells, which showed similar localization as the infiltrating CD11c+ cells, were seen to aggregate in the central area of the spongiosis in these cases. In addition, infiltrating CD4+ cells, the numbers of which varied greatly, were also observed in the central area of the spongiosis in the five cases, in agreement with the localization of IgE+ CD11c+ cells. Meanwhile, infiltrating CD8+ cells and CD207+ cells showed a tendency to localize around them in the peripheral areas of the spongiosis (Figure 4: case 1). In the analysis using paraffin-embedded sections, the levels of Der-f1 antigens were below the detection limit, and the levels of Mite Extract antigens were insufficient for sensitivity and specificity calculations.

In the comparisons between the AD and control cases in the mean numbers of CD207+ cells in the epidermis and CD11c+ cells both in the epidermis and upper dermis, only the mean numbers of CD11c+ cells in the epidermis (i.e., IDECs) were significantly higher in the AD cases than in the control cases with non-eczematous inflammatory skin disorders and serum hyper-IgE (Table 2).

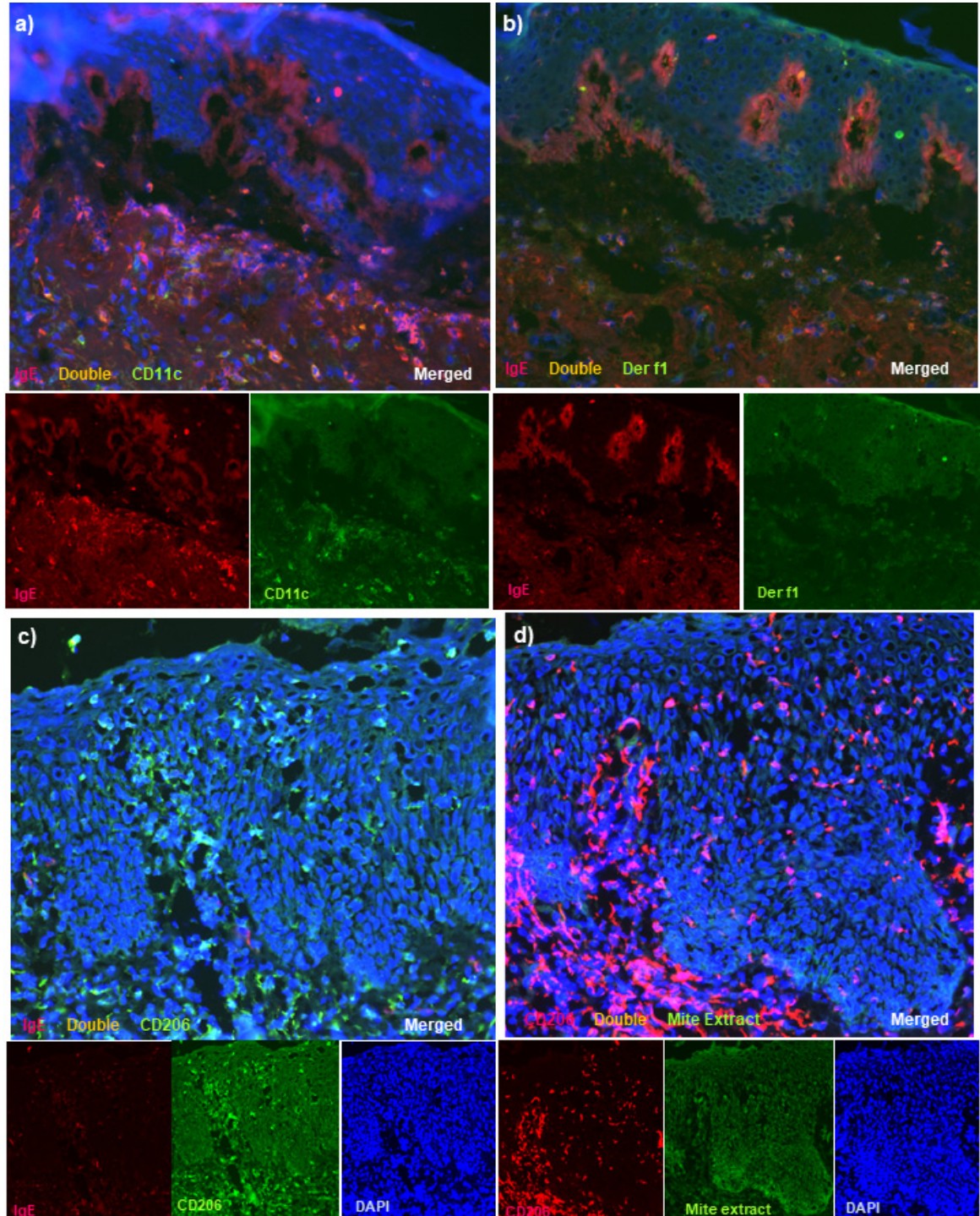

**Figure 3.** Double-immunofluorescence staining for skin lesions in control subjects. (**a**) In a control case of non-eczematous inflammatory skin disorders and serum hyper-IgE (bullous pemphigoid), double-positive IgE+ CD11c+ cells (yellow) were seen in small numbers among the infiltrated cells in the upper dermis, and they were scarcely detected in the epidermis. (**b**) In the same control case, double-positive IgE+ Der f1+ cells were not observed in the epidermis or the upper dermis. Note: in (**a**,**b**), linear depositions of IgE at the basal layer (red) were characteristically observed. (**c**) In a control case of inflammatory skin disorders and spongiotic tissue within the epidermis (nummular eczema), single-positive IgE-CD206+ cells (green) were observed in the spongiotic epidermis. (**d**) In the same control case, infiltrating CD206+ cells were scarcely detected with Mite Extract antigens in the spongiotic epidermis. (**a**,**b**): Case 11. (**c**,**d**): Case 13. Note: in (**a**,**b**), the single-staining images of DAPI were omitted from the figures. Sets of figures, i.e., (**a**–**d**), represent serial sections. Original magnifications: 200×, (**a**–**d**).

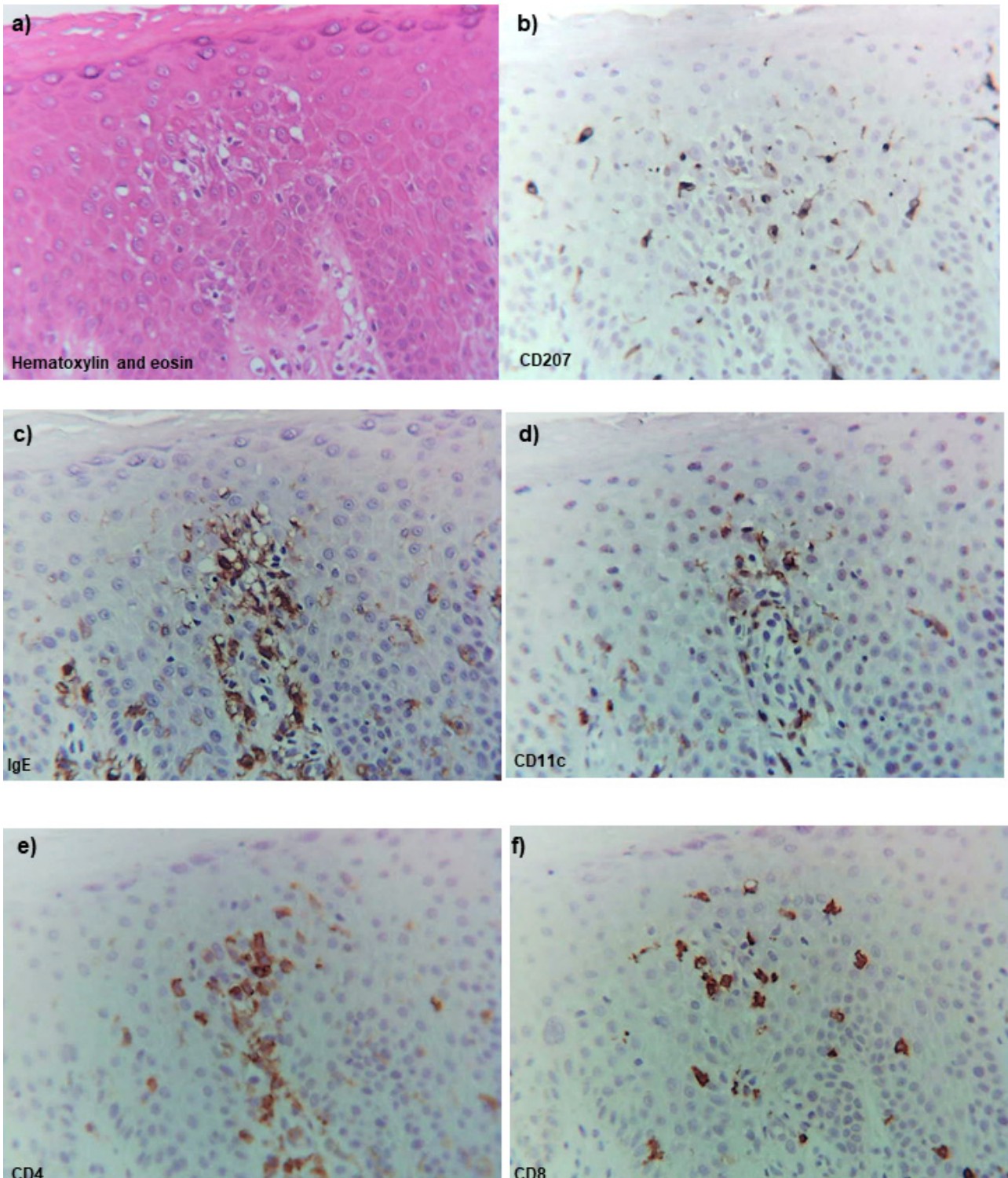

**Figure 4.** Routine and single-immunohistochemical staining for active lesions of chronic AD in a patient with IgE-allergic AD and HDM allergy. (**a**) An early lesion of spongiosis with the infiltration of mononuclear cells in the epidermis. (**b**) CD207+ cells were mainly seen in the peripheral area of the spongiosis. (**c**) Infiltrating IgE+ cells were observed to aggregate in the central area of the spongiosis. (**d**) Infiltrating CD11c+ cells that show similar localization as the infiltrating IgE+ cells were observed in the central area of the spongiosis. (**e**) Infiltrating CD4+ cells that showed similar localization as the infiltrating IgE+ cells and CD11c+ cells were observed in the central area of the spongiosis. (**f**) Infiltrating CD8+ cells were mainly seen in the peripheral area of the spongiosis. (**a**): Case 1; hematoxylin-eosin staining. (**b–f**): Case 1; single-immunohistochemical staining. Sets of (**a–f**) represent serial sections. Original magnification: 200×, (**a–f**).

## 4. Discussion

Although the pathogenetic roles of HDMs (mainly D. farinae and D. pteronyssinus) in AD remain controversial [3], clinical studies have demonstrated that the prevalence of serum-specific IgEs against HDM allergens, the major allergens for patients with IgE-allergic AD, is approximately 70% to 95% in patients with IgE-allergic AD [17,18]. It has also been reported that the prevalence of having HDMs on the skin is 47.3% in patients with AD, while the prevalence of having the major HDM antigens (Der p1 and Der f1) on the skin surface is 93.9% in normal healthy volunteers [19,20]. In addition, the pathogenic characteristics of HDMs that induce eczematous lesions in the normal skin of AD patients have been used in atopy patch testing with a 40% to 50% provocation-positive rate [21,22]. In a previous immunohistopathological study before the concept of IDECs was known, the presence of HDM antigens localized with LCs had been demonstrated in naturally occurring AD lesions in 19 of 31 (61.3%) patients with IgE-allergic AD and allergic sensitization to HDMs [23]. The study also showed that seven AD patients with negative results for serum specific IgEs against HDMs were all negative for tissue HDM antigens in the lesioned skin [23]. In recent studies, the distinct behavior and localization patterns of LCs and IDECs in lesioned AD skin have been revealed: activated LCs are increased in the areas underneath the TJs, and the LCs extend their dendrites through the TJs, likely to capture antigens from the outside of the TJ barrier, while IDECs localize to the lower area of the epidermis [24]. On the other hand, in our preliminary report, we demonstrated that IgE-bearing IDECs could infiltrate into the central area of spongiosis in naturally occurring skin lesions of IgE-allergic AD patients. In addition, we also demonstrated that HDM antigen (Der f1)-loaded IgE-bearing LCs could present in peripheral areas of the spongiosis or in areas of the non-spongiotic epidermis, such as below TJs [13].

In the present immunohistopathological studies with double-immunofluorescence staining techniques for adult and elderly patients with IgE-allergic AD and HDM allergy, we confirmed that along with T-cell infiltration, IgE-bearing IDECs (CD11c+ and CD206+ cells) infiltrated and aggregated in the central area of the spongiotic epidermis in active lesions of chronic AD [25] in all of the AD patients [13]. However, in the non-spongiotic epidermis, they were scattered in the middle to lower epidermis [13,14]. In addition, we demonstrated the presence of IgE-bearing IDECs coexisting with HDM antigens (i.e., Der f1 and/or Mite Extract antigens) gathered in the spongiotic epidermis of skin lesions in four of the six (66.7%) AD patients (Figures 1 and 2). Furthermore, LCs (CD207+ cells; probably IgE-bearing CD207+ cells [13]) loaded with HDM antigens were also observed in the peripheral areas of the spongiosis. Analysis of the single staining of serial paraffin-embedded sections showed that in the AD cases with the coexistence of IgE-bearing IDECs and HDM antigens in the aforementioned analysis, infiltrating CD4+ T cells coexisting with IgE+ CD11c+ IDECs were present in the central area of the spongiosis in association with the infiltration of CD8+ T cells and CD207+ LC cells in the surroundings (Figure 4).

Although it could not be analyzed, we presume that the IgE-bearing IDECs and LCs that have HDM antigens might express some cytokines in the epidermis, or they might move to the dermis and lymph nodes to present the HDM antigens to T cells [22,26]. Furthermore, from the results of the above-mentioned morphological analysis, we speculate that IgE-bearing IDECs and LCs may present HDM antigens to T cells even in the spongiotic epidermis.

In the analysis of cell infiltration in the upper dermis, infiltrating double-positive IgE+ CD11c+ cells (i.e., IgE-bearing dermal inflammatory DCs) were mainly observed in the papillary and subpapillary dermis in the lesioned skin of patients with IgE-allergic AD; the numbers were significantly higher in patients with IgE-allergic AD than in the control subjects with non-eczematous inflammatory skin disorders and serum hyper-IgE.

In the control cases, results of double-immunofluorescence staining indicated the following: IgE-bearing IDECs and those with HDM antigens were hardly detected in the epidermis of the skin lesions of patients with non-eczematous inflammatory skin disorders and serum hyper-IgE, and in the epidermis of the skin lesions of patients with inflammatory

skin disorders and spongiotic tissue within the epidermis, IDECs with HDM antigens were scarcely detected in the spongiotic epidermis in which IDECs without IgE had sufficiently infiltrated (Figure 3a–d; Table 2).

Taken together, we speculate that in IgE-allergic AD with HDM allergy, IgE-bearing IDECs responding to the HDM antigens may play a major role in the formation of the characteristic eczematous reaction, i.e., spongiotic formation with the infiltration of CD4+ and CD8+ T cells; in addition, these T cells may be capable of expressing cytokines, such as interferon-gamma, Fas ligand, perforin, and granzyme B, which induce apoptosis in keratinocytes in the epidermis [26–29]. However, LCs might be the first immune-reactive cells that trigger a response against HDM antigens to induce the infiltration of the IgE-bearing IDECs into the pre-spongiotic epidermis [26]. We consider that these immunoreactions may reflect IgE-mediated delayed-type hypersensitivity [8]. We suppose that one of the major differences between AD and other eczematous disorders (e.g., contact dermatitis) that show similar tissue reactions, such as the accumulation of IDECs in the spongiotic epidermis [30], is whether IgEs are involved in the eczematous reactions.

There are some limitations in this study, including the inherent technical limitations of immunohistopathological analyses, the lack of a blind method for quantitative evaluations, and the small sample sizes of the AD and control cases. In addition, since the present analysis focused on the role of IgEs, we did not include patients with non-IgE-allergic (intrinsic form) AD or enough numbers of patients with other types of eczematous disorders with spongiotic epidermis (e.g., allergic contact dermatitis) as study and control subjects. We consider that patients with these disease conditions should be analyzed in a future study.

## 5. Conclusions

Nevertheless, even considering such limitations, we consider that the results of the present study demonstrated the crucial role of HDM allergens in the immunopathogenesis of eczematous dermatitis in IgE-allergic AD, in which the IgE-mediated delayed-type hypersensitivity reaction, together with IgE-bearing DCs (i.e., IDECs, LCs, and dermal inflammatory DCs), specific T cells, keratinocytes, and HDM antigens, may lead to spongiosis formation.

**Author Contributions:** Conceptualization, R.T.; methodology, R.T. and Y.H.; software, R.T.; validation, R.T. and Y.H.; formal analysis, R.T. and Y.H.; investigation, R.T. and Y.H.; resources, R.T.; data curation, R.T.; writing—original draft preparation, R.T.; writing—review and editing, R.T. and Y.H.; visualization, R.T. and Y.H.; supervision, R.T.; project administration, R.T.; funding acquisition, R.T. All authors have read and agreed to the published version of the manuscript.

**Funding:** This research received no external funding.

**Institutional Review Board Statement:** The study was conducted according to the guidelines of the Declaration of Helsinki and approved by the Ethics Committee of the Tokyo Metropolitan Geriatric Hospital and Institute of Gerontology (No. R15-42, 9 December 2016).

**Informed Consent Statement:** Subjects in this study all provided written informed consent for each biopsy and for the use of their specimens for research.

**Conflicts of Interest:** The authors declare no conflict of interest.

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
