# Peer review of "Immunohistopathological Analysis of Immunoglobulin E-Positive Epidermal Dendritic Cells with House Dust Mite Antigens in Naturally Occurring Skin Lesions of Adult and Elderly Patients with Atopic Dermatitis"

_dermatopathology, doi:10.3390/dermatopathology8030045_

Round 1

Reviewer 1 Report

It is an interesting study but some changes are needed:

In the abstract it is said that control were patients with inflammatory skin disorders. In the material and methods it is said that some controls didn’t have inflammatory skin disordes and other controls have. Please clarify and explain why you chose patients with inflammatory skin disorders as controls instead of healthy participants.

In AD cases and 73 the control cases with non-eczematous inflammatory skin disorders, no statistically sig-74 nificant difference was found in mean value ± standard deviation of serum total IgE (AD 75 cases: 33822.8 ± 45077.2 IU/ml, control cases: 27982.2 ± 49632.9 IU/ml; p = 0.835). This sentence should be included in the results section

Clinical and laboratory data of the AD patients and control subjects are summa-84 rized in Table 1. This sentence and this table should be included in the results

Similar to our previous report [17]. You can use this sentence in the discussion but you should only provide your results without comparisons with other reports in the result section.

You could do some comparison in HE and IF between patients and control providing the mean (SD) and p values

Author Response

To Reviewer 1

We greatly appreciate your positive evaluation of our article.

Based on your suggestions, we have revised our manuscript as follows:

Your comment: In the abstract it is said that control were patients with inflammatory skin disorders. In the material and methods, it is said that some controls didn’t have inflammatory skin disorders and other controls have. Please clarify and explain why you chose patients with inflammatory skin disorders as controls instead of healthy participants.

Our reply: In our article, the control subjects all had some kind of inflammatory skin disorders (with serum hyper IgE or spongiotic tissue within the epidermis).

In the material and methods, we added

“These control subjects were selected to confirm that the localization of IgE-positive DCs and mite antigens in eczematous dermatitis of IgE-allergic AD could be characteristic findings of AD compared to other inflammatory skin disorders. We did not add healthy participants to the control subjects, because our preliminary analysis did not reveal any specific findings in their skin.”

Your comment: In AD cases and the control cases with non-eczematous inflammatory skin disorders, no statistically sig-nificant difference was found in mean value ± standard deviation of serum total IgE (AD 75 cases: 33822.8 ± 45077.2 IU/ml, control cases: 27982.2 ± 49632.9 IU/ml; p = 0.835). This sentence should be included in the results section.

Our reply: This sentence has moved into the results section.

Your comment: Clinical and laboratory data of the AD patients and control subjects are summarized in Table 1. This sentence and this table should be included in the results.

Our reply: The sentence and Table 1 have moved into the results section, and two subtitles have been added in the results section.

Your comment: Similar to our previous report [17]. You can use this sentence in the discussion but you should only provide your results without comparisons with other reports in the result section.

Our reply: This sentence “Similar to our previous report [17]” and cited reference (17) have been deleted.

Your comment: You could do some comparison in HE and IF between patients and control providing the mean (SD) and p values.

Our reply: We have selected the comparison items to focus on the research theme in this article. I would appreciate your understanding.

Additional fixes:

We made a mistake in a description of the statistical method used, so please let me correct it from “the Mann-Whitney U test” to “the Welch test” (line 130).

Reviewer 2 Report

This is a very well written manuscript describing a well done study.

The authors acknowledge a major limitation, the lack of a blinded assessment. It may be good to add that limitation to the abstract.

Author Response

To Reviewer 2

We greatly appreciate your positive evaluation of our article.

Due to the limited number of characters in the abstract, we did not add the major limitation and the lack of a blinded assessment of our study in the abstract of our revised article. I would appreciate your understanding.

In addition, based on suggestions of Reviewer 1, we have revised our manuscript as follows:

Comment of Reviewer 1: In the abstract it is said that control were patients with inflammatory skin disorders. In the material and methods, it is said that some controls didn’t have inflammatory skin disorders and other controls have. Please clarify and explain why you chose patients with inflammatory skin disorders as controls instead of healthy participants.
Our reply: In our article, the control subjects all had some kind of inflammatory skin disorders (with serum hyper IgE or spongiotic tissue within the epidermis).
In the material and methods, we added 
“These control subjects were selected to confirm that the localization of IgE-positive DCs and mite antigens in eczematous dermatitis of IgE-allergic AD could be characteristic findings of AD compared to other inflammatory skin disorders. We did not add healthy participants to the control subjects, because our preliminary analysis did not reveal any specific findings in their skin.”

Comment of Reviewer 1: In AD cases and the control cases with non-eczematous inflammatory skin disorders, no statistically sig-nificant difference was found in mean value ± standard deviation of serum total IgE (AD 75 cases: 33822.8 ± 45077.2 IU/ml, control cases: 27982.2 ± 49632.9 IU/ml; p = 0.835). This sentence should be included in the results section.
Our reply: This sentence has moved into the results section.

Comment of Reviewer 1: Clinical and laboratory data of the AD patients and control subjects are summarized in Table 1. This sentence and this table should be included in the results.
Our reply: The sentence and Table 1 have moved into the results section, and two subtitles have been added in the results section.

Comment of Reviewer 1: Similar to our previous report [17]. You can use this sentence in the discussion but you should only provide your results without comparisons with other reports in the result section.
Our reply: This sentence “Similar to our previous report [17]” and cited reference (17) have been deleted. 

Comment of Reviewer 1: You could do some comparison in HE and IF between patients and control providing the mean (SD) and p values.
Our reply: We have selected the comparison items to focus on the research theme in this article. I would appreciate your understanding.

Additional fixes:
We made a mistake in a description of the statistical method used, so please let me correct it from “the Mann-Whitney U test” to “the Welch test” (line 130).